# Mott Cell Differentiation in Canine Multicentric B Cell Lymphoma with Cross-Lineage Rearrangement and Lineage Infidelity in a Dog

**DOI:** 10.3390/vetsci9100549

**Published:** 2022-10-05

**Authors:** Woo-Sub Kim, Kun-Ho Song, Hyeona Bae, DoHyeon Yu, Joong-Hyun Song

**Affiliations:** 1Department of Veterinary Internal Medicine, College of Veterinary Medicine, Chungnam National University, Daejeon 34134, Korea; 2College of Veterinary Medicine, Gyeongsang National University, Jinju 52828, Korea

**Keywords:** cross-lineage rearrangement, dog, flow cytometry, lineage infidelity, lymphoma, Mott cell, PCR for antigen receptor rearrangement

## Abstract

**Simple Summary:**

The scientific literature regarding Mott cell differentiation in canine lymphoma is scarce. Mott cells are defective in immunoglobulin secretion and are derived from plasma cells, and lymphoma is a severe condition characterized by the proliferation of neoplastic lymphoid cells. Lymphoma can be divided into B- or T-cell according to their origin. Whether the origin of lymphoma is B- or T-cell can be confirmed by PCR for antigen receptor rearrangement or flow cytometry assay. However, the phenomenon in which B- and T-cells are simultaneously identified in PCR for antigen receptor rearrangement and flow cytometry is called cross-lineage rearrangement and lineage infidelity, respectively, and is known to be occasionally found in canine lymphoma. These phenomena have not been reported in canine lymphoma with Mott cell differentiation. This study is the first report of Mott cell differentiation in canine B-cell lymphoma with cross-lineage rearrangement and lineage infidelity. This study describes the clinical features, diagnosis, and treatment of this unknown type of cancer in a 4-year-old female mongrel dog.

**Abstract:**

Lymphoma is a severe condition characterized by the proliferation of neoplastic lymphoid cells. A 4-year-old female mongrel dog presented with solitary lymph node enlargement. Significant right prescapular lymphadenopathy and abdominal enlargement were observed during physical examination. A complete blood count revealed lymphocytosis, and a peripheral blood smear revealed lymphoblastosis and Mott cells. Fine needle aspiration cytology (FNAC) of the right prescapular lymph node revealed a predominant population of lymphoblasts and Mott cells. Based on the FNAC and blood smear results, the patient was diagnosed with leukemic state multicentric B-cell lymphoma with Mott cell differentiation. Subsequent PCR for antigen receptor rearrangement and flow cytometry revealed that the patient exhibited cross-lineage rearrangement (CLRA) and lineage infidelity (LI), respectively. CHOP-based chemotherapy was initiated, however, the patient’s disease was progressive. The patient died three months after the initial presentation. Mott cell differentiation in canine B-cell lymphoma (MCL) has rarely been reported in the veterinary literature and seems to show an unusual clinical course. To the best of our knowledge, no reports of MCL with CLRA and LI exist. We report the clinical features, diagnosis, and treatment of MCL with CLRA and LI.

## 1. Introduction

Lymphoma (lymphosarcoma, LSA) is the uncontrolled proliferation of neoplastic lymphoid cells that arises in lymphoid- or other tissues in the body. It is commonly encountered in small animal practices and comprises approximately 83% of canine hematopoietic cancers [1]. No sex predisposition has been reported. LSA can occur at any age but is most common in middle-aged to older dogs. One study reported an age-adjusted overall incidence of 1.5/100,000 for dogs <1 year of age and 84/100,000 for 10-year-old dogs [2]. Although the etiology of canine LSA is poorly understood, it is likely multifactorial (e.g., genetic, molecular, environmental, immunological, and infectious factors) [3]. LSA can be classified based on anatomic location (e.g., multicentric, gastrointestinal, mediastinal, and cutaneous LSA), histopathologic features, immunophenotypic characteristics (i.e., B- and T-cell LSA), and World Health Organization (WHO) clinical staging system [3,4,5]. The diagnosis of LSA usually depends on morphological characteristics identified using fine-needle aspiration cytology (FNAC) and/or histopathology. Additionally, the presence of lymphoblasts exceeding 50% is often used as a diagnostic hallmark of LSA [4]. In addition to these tests, additional diagnostics such as immunohistochemistry (IHC), flow cytometry (FC), and PCR for antigen receptor rearrangement (PARR) have been developed to assist in the diagnosis and classification of LSA. As canine LSA is systemically affected in most patients, chemotherapy is the mainstay of treatment. Despite numerous chemotherapy protocols being available, CHOP-based chemotherapy (i.e., cyclophosphamide [C], doxorubicin [hydroxydaunorubicin], vincristine [oncovin], and prednisone/prednisolone [P]) is considered the most effective treatment protocols for canine LSA [2,6]. LSA prognosis varies depending on numerous factors, such as the stage, location, clinical remission in response to chemotherapy, immunophenotypes, and characteristics of the neoplastic cells [7,8].

Mott cells are defective in immunoglobulin secretion and are derived from plasma cells [9,10,11]. The multiple spherical inclusions of Mott cell cytoplasm represent immunoglobulin (Ig) accumulation in the rough endoplasmic reticulum, called the Russel body [9,10,11]. Mott cell differentiation is associated with pathological conditions in some diseases, including chronic inflammation, autoimmune diseases, multiple myeloma, plasma cell dyscrasias, and LSA [11]. Since Mott cells are differentiated B cells, they can appear in B-cell LSA [12]. To the best of our knowledge, nine reports of Mott cell differentiation in canine B-cell lymphoma (MCL) have been written to date [13,14,15,16,17,18,19,20,21]. The exact etiology and clinical course of MCL are poorly understood. Since most patients with MCL in previous studies were euthanized at the time of diagnosis or after a short course of chemotherapy, data on MCL regarding treatment, response to chemotherapy, and prognosis are lacking. MCL is rare and reportedly has an unusual clinical course [13]. Therefore, investigating various MCL cases that have studied the clinical effectiveness of chemotherapy and prognosis is important. Furthermore, MCL with cross-lineage rearrangement (CLRA) or lineage infidelity (LI) has not been reported to date, and its response to chemotherapy and clinical course are entirely unknown. Herein, we report a case of MCL with CLRA and LI, along with the clinical features, diagnosis, and treatment thereof.

## 2. Case Presentation

A 4-year-old female mongrel dog presented with solitary lymph node enlargement. There was no historical evidence of toxicant exposure or infection. Physical examination revealed significant right prescapular lymphadenopathy (3.9 × 3.5 cm^2^) and abdominal enlargement. Complete blood count (CBC, ADVIA^®^ 2120, Siemens Healthcare Diagnostics, Deerfield, IL, USA) revealed severe lymphocytic leukocytosis (lymphocyte: 16,390 cells/μL [reference interval: 1300–4100 cells/μL]; leukocyte: 28,850 cells/μL [reference interval: 5200–13,900 cells/μL]) and mild anemia (hematocrit: 30.9% [reference interval: 37.1–57.0%]) (Table 1). A blood smear revealed abundant lymphoid cells and the appearance of Mott cells (Figure 1).

The results of serum biochemistry (BS-200 Chemistry Analyzer; MINDRAY^TM^, Shenzhen, China) were unremarkable. Abdominal ultrasonography detected prominent intra-abdominal lymphadenopathy, ascites, and a splenic mass with a honeycomb sign (Figure 2). Fine needle aspiration cytology (FNAC) of the right prescapular lymph node revealed predominant lymphoblasts and several Mott cells (Figure 1).

The serosanguineous ascites were identified as exudates (total nucleated cell count: 148,300 cells/µL and total protein: 4.1 g/dL) using CBC and a refractometer, and cytology revealed predominant lymphoid cells, consistent with the features of neoplastic effusion (Figure 3).

A PARR (IDEXX Laboratories, Westbrook, ME, USA) assay using the right prescapular lymph node (LN) revealed that the patient presented with clonal rearrangement of both IG and T-cell receptor genes, a phenomenon known as CLRA.

Immunophenotyping via FC assay was performed on right prescapular LN aspirates and peripheral blood samples. In order to select the best gating strategy for the determination of the lymphoid cells in the right prescapular LN and peripheral blood, lymphoid cells were identified and gated by forward scatter (FSC) and side scatter (SSC) characteristics. After doublet exclusion of the gated lymphoid cells, the singlets were gated (Figure 4). Then, immunophenotyping was performed on gated singlet lymphoid cells of the LN and peripheral blood using CD3, CD5, CD21, and CD34 antibodies. Lymphoid cells from the LN and peripheral blood showed a homogenous population (approximately 90%) of CD3+/CD5−/CD21+/CD34− and CD3−/CD5−/CD21+/CD34−, respectively (Table 2, Figure 4). Since CD3 represents the T-cell marker and CD21 represents the B-cell marker, a phenomenon known as LI was found on the right prescapular LN.

Based on the results of FNAC, PARR, FC, and the presence of Mott cells, the patient was diagnosed with stage Va (i.e., leukemic state) multicentric B-cell LSA with Mott cell differentiation, CLRA, and LI. CHOP-based chemotherapy using cyclophosphamide (250 mg/m^2^, IV), vincristine (0.7 mg/m^2^, IV), doxorubicin (1 mg/kg, IV), and prednisolone (2.0 mg/kg, PO, q24h for seven days, then slowly tapered until the final visit), was initiated. The patient was monitored weekly for clinical response to chemotherapy throughout the protocol, and abdominal ultrasonography was repeated at the end of each chemotherapy cycle.

The initial response to chemotherapy was favorable during the first cycle of CHOP therapy. The enlarged superficial and intra-abdominal LNs decreased in size by approximately 40%, and lymphocytosis on CBC was within the reference interval (4.05 × 10^9^ cells/L).

However, complete remission (CR) was not achieved, and partial remission (PR) was maintained for the first four weeks. In the 6th week of chemotherapy (i.e., on the first day of the second cycle), the patient’s superficial and intra-abdominal LNs and spleen were observed to be enlarged, which was considered a progressive disease (PD), followed by CHOP chemotherapy and L-asparaginase administration (400 U/kg, SC). The response to re-instituted chemotherapy over four weeks was consistently poor, and palliative therapy using chlorambucil (6 mg/m^2^, PO, q24h) and prednisolone (1.0 mg/kg, PO, q24h) was initiated after L-CHOP chemotherapy at the request of the owner. Unfortunately, the response to this treatment was also consistently poor, and superficial and intra-abdominal LNs and the spleen gradually increased in size until the final visit (Figure 5). The patient eventually became anorexic and lethargic three months after the initial presentation and expired (survival time of 81 days after the initial treatment).

## 3. Discussion

In the present report, we describe the diagnostic and therapeutic trials of a dog with multicentric B-cell LSA, as well as Mott cell differentiation, CLRA, and LI. Although histopathological examination could not be performed via IHC assay due to the owner’s refusal, we were finally able to diagnose the patient with MCL based on the results of FNAC (existence of lymphoblastosis and Mott cell differentiation) and FC analysis of peripheral blood (immunophenotyping of CD3−/CD5−/CD21+/CD34−). Mott cells are derived from terminally differentiated B cells, and neoplastic lymphoid differentiation combined with Mott cell differentiation is indicative of B-cell LSA [12]. These results suggest multicentric B-cell LSA with Mott cell differentiation, CLRA, and LI in this patient. In addition, numerous lymphoblasts and Mott cells were found in the patient’s peripheral blood smears. Lymphoblasts in the peripheral blood are representative signs of leukemic LSA or leukemia. It is difficult to differentiate between LSA and leukemia based on the presence of lymphoblasts in the blood alone. LSA can be differentiated from leukemia with clinical findings such as the presence of generalized lymphadenopathy and flow cytometric analysis of CD34. CD34 is a transmembrane phosphoglycoprotein encoded by CD34 in various species, including humans and dogs [22]. CD34-expressing cells are normally found in the umbilical cord and bone marrow as hematopoietic cells [23]. Because of these characteristics, CD34 is often used clinically to differentiate leukemic state LSA from acute lymphocytic leukemia (ALL) and ALL from chronic lymphocytic leukemia (CLL), although it is not always definitive [4]. Since both CD34+ LSA and CD34− leukemia have been reported in canine patients, CD34 alone cannot completely rule out leukemia. However, because generalized lymphadenopathy and predominant CD34^−^ lymphoid cells were identified in the present patient, it was considered that the likelihood of leukemic state lymphoma was high due to the following characteristics of the patient.

In the present patient, CLRA and LI were found using PARR and FC, respectively. PARR is a methodology used to detect clonality in B-cell and T-cell LSA [24]. Normally, Ig or T-cell receptor (TCR) gene rearrangements are considered to be of B- or T-lineage origin, respectively. However, CLRA, a phenomenon that breaks this view, is a finding comprising cross-lineage expression of lineage-specific immunological markers on B- or T-cells (i.e., both Ig and TCR on PARR) [25,26,27,28,29]. One study previously reported this uncommon phenomenon in 21% of dogs with marginal zone lymphoma and 5% of dogs with T-cell LSA [27,30]. Unfortunately, the precise mechanism of CLRA in LSA is poorly documented, and its effects on prognosis or response to treatment have not been determined. Further studies on the clinical implications of CLRA are warranted.

FC is a useful test for diagnosing immunophenotype LSA and has the advantage of providing a larger panel of markers than IHC or ICC [4]. It is highly sensitive and specific for diagnosing LSA, has 94% agreement with IHC, and is superior to the PARR test [31]. In the FC assay of the right prescapular LN, most lymphocytes expressed CD3+/CD5−/CD21+/CD34−. CD3 is a T cell co-receptor expressed in the membranes of normal and neoplastic T cells [32]. Because CD3 is present at all stages of T-cell development, it is a useful marker for identifying T-cell LSA. Likewise, CD21, which is a protein expressed on B cells, is used to identify B-cell LSA [33]. However, the patient’s lymphocytes in the right prescapular lymph node showed both CD3 and CD21, making it difficult to determine the phenotype. This phenomenon, known as LI, causes chemoresistance in LSA and leukemia in humans and is considered a negative prognostic factor [29,34,35,36,37,38]. Although the cause of this phenomenon is unclear, chemoresistance may be related to the high prevalence of drug efflux pump expression and the high proportion of cytogenetic abnormalities [36,37,38,39]. Several reports in humans regarding the aberrant expression of CD markers in LSA and leukemia have been published to date. However, the clinical implications of LI in canine LSA are poorly understood. A previous study reported the lineage differentiation of canine LSA using FC [28]. Of the 59 dogs, 13 had LI. In this study, leukemic states occurred in all three phenotypes (i.e., B-, T-cell, and LI); however, LI cases comprised the largest proportion. Similarly, our patient was also in a leukemic state. LI may be related to leukemic LSA or leukemia. Since FC is useful in diagnosing and evaluating LSA prognosis, it should be considered as a front-line test in patients with LSA, and further studies on the clinical implications of LI in canine LSA are needed.

Although rarely curable, most LSAs are initially responsive to treatment, and LSA that responds to chemotherapy has a better prognosis than LSA that does not [7,8,40]. In previous studies, most patients with MCL were euthanized at the time of diagnosis, but some received chemotherapy (e.g., COP and CHOP and sole prednisone) [14,15,16,17,18,20,21]. There are only two cases in which MCL was treated using CHOP [16,17]. In one study, an MCL dog treated with CHOP improved clinically in early induction [16]. However, the patient became ill at week 10 of chemotherapy, and the dog was euthanized after three months of initial presentation [16]. In another study, an MCL dog treated with CHOP also achieved clinical remission initially, but recurred 2.5 months after induction [17]. Although rescue chemotherapy was attempted multiple times, the patient showed a partial and temporary response only to dacarbazine [17]. The dog was euthanized due to severe and uncontrolled seizures nine months after the initial diagnosis [17]. Likewise, After the diagnosis, our patient received CHOP-based chemotherapy, including L-asparaginase. However, the response to chemotherapy was transient and poor. The patient died three months after the initial presentation. This is significantly different compared to the 10–14-month mean survival time (MST) of multicentric B-cell LSA [41]. Furthermore, most patients with MCL have extensive nodal and extranodal involvement [13]. Likewise, the current patient showed a poor response to chemotherapy and had extensive nodal and extranodal involvement, including the spleen, superficial and abdominal lymph nodes, and bone marrow at the initial presentation. Although splenectomy, bone marrow examination, and necropsy, which are tests that can confirm LSA infiltration, were not performed due to the owner’s refusal, lymphoblastic abdominal effusions around the spleen and lymphoblasts on peripheral blood smears suggested the involvement of the spleen and bone marrow, respectively. Extrapolation of MCL prognosis is difficult due to the small number of reports; however, it appears that MCL is considered a negative prognostic factor in canine LSA. In addition to MCL as an important prognostic factor, CLRA expression in PARR assays and atypical immunophenotypic features of LI should be investigated in canine LSA. However, it is beyond the scope of this report, and further large-scale retrospective reviews are needed. Moreover, to broaden our knowledge of MCL with CLRA and LI and to establish tailored treatment strategies, a complete characterization of this type of LSA is required, as is further investigation of other case series and studies about the clinical implications of CLRA and LI.

## 4. Conclusions

To our knowledge, this is the first case report of MCL with CLRA and LI. The patient’s response to therapy remained consistently poor despite treatment with CHOP-based chemotherapy. Taken together, MCL might be considered a negative prognostic factor and appears to have an unusual clinical course in canine B-cell LSA. A complete characterization of this type of LSA requires further investigation with additional case studies, and further studies on the clinical implications of CLRA and LI are required.

## Figures and Tables

**Figure 1 vetsci-09-00549-f001:**
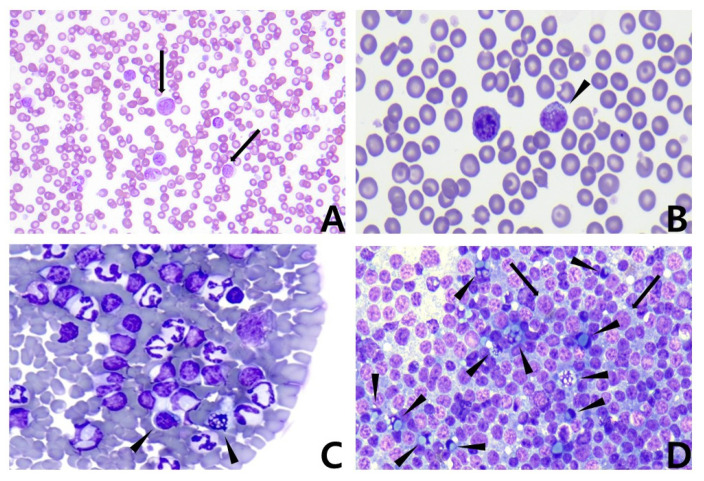
Microscopic images. Modified Wright’s stain. ×40 objective; (**A**) Peripheral blood smear. Arrows indicate lymphoblasts. (**B**) Peripheral blood smear. Arrowhead indicates Mott cell. (**C**) Peripheral blood smear. Arrowheads indicate Mott cells. (**D**) Fine-needle aspiration cytology (FNAC) of the right prescapular lymph node. Arrowheads indicate Mott cells, whereas arrows indicate lymphoblasts. Note the predominant population of lymphoblasts and its difference in size compared with small lymphocytes.

**Figure 2 vetsci-09-00549-f002:**
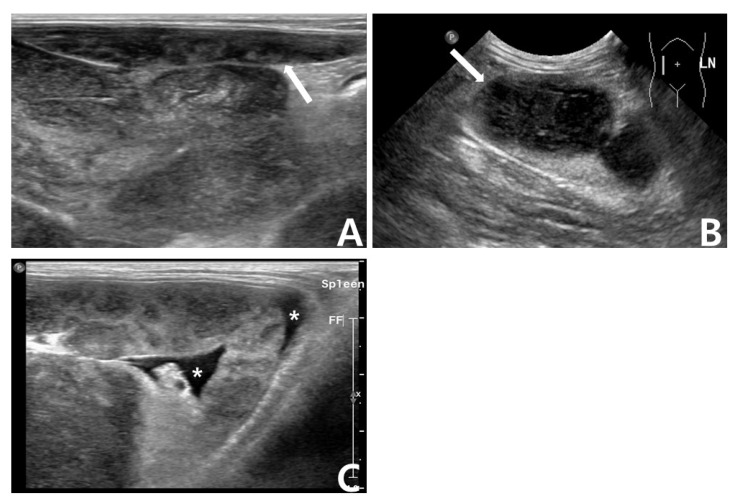
Abdominal ultrasonography. (**A**) Honeycomb-like lesion of splenic parenchyma. The arrow indicates the spleen. (**B**) Enlarged mesenteric lymph nodes. The arrow indicates enlarged lymph nodes. (**C**) Ascites around the spleen. The asterisks indicate ascites around the spleen.

**Figure 3 vetsci-09-00549-f003:**
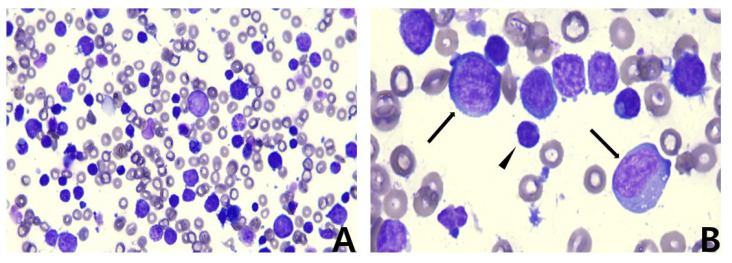
Microscopic images of ascites. Modified Wright’s stain. (**A**) A predominant population of lymphoid cells was found in a high-power field. ×40 objective. (**B**) Arrows indicate lymphoblasts and the arrowhead indicates normal small lymphocytes. ×100 objective. Note the predominant lymphoblast population and its difference in size compared with that of the small lymphocytes.

**Figure 4 vetsci-09-00549-f004:**
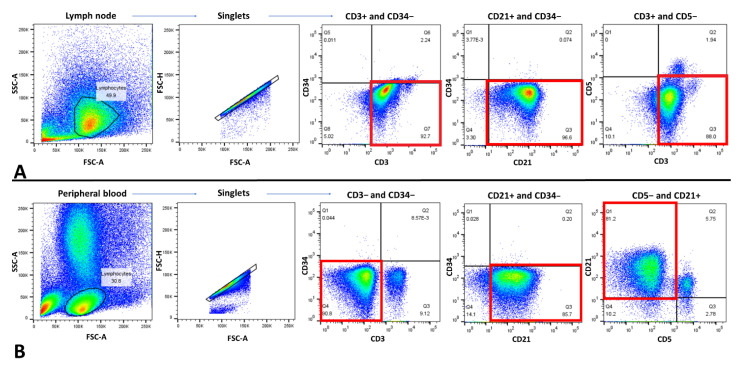
Flow cytometry (FC) of the right prescapular lymph node (LN) and peripheral blood (PB). Forward versus side scatter plots show the placement of the lymphoid cell gate in right prescapular LN and PB. The cells shown in two-parameter density plots with four quadrants were gated on the singlet lymphoid cell populations after doublet exclusion. (**A**) Right prescapular LN. Note that approximately 90% of lymphoid cells in the right prescapular LN show simultaneous CD3 and CD21 positivity (red squares). (**B**) PB. Note that approximately 90% of lymphoid cells in the PB only show CD21 positivity (red squares).

**Figure 5 vetsci-09-00549-f005:**
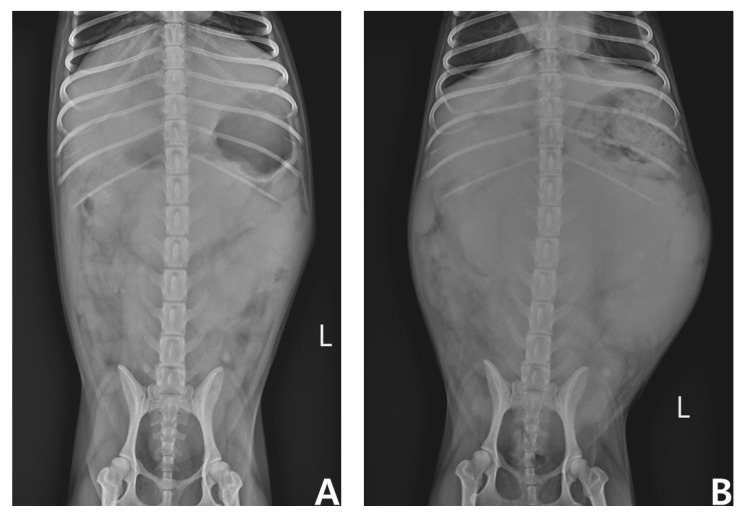
Ventrodorsal abdominal radiographs at the first and final visits (74 days). (**A**) First visit. (**B**) Final visit. Note the significant difference in abdomen size between A and B due to the enlarged spleen.

**Table 1 vetsci-09-00549-t001:** Complete blood count results of the patient.

Parameters	Reference Interval	Results
RBC (×1012/L)	5.7–8.8	4.4
Hct (%)	37.1–57.0	30.9
Hgb (g/dL)	12.9–18.4	9.79
WBC (×10^9^/L)	5.20–13.90	28.85
Neutrophils (×10^9^/L)	3.90–8.00	9.42
Lymphocytes (×10^9^/L)	1.30–4.10	16.39
Monocytes (×10^9^/L)	0.20–1.10	2.59
Eosinophils (×10^9^/L)	0.00–0.60	0.08
Platelets (×10^9^/L)	143–400	370

Hct, hematocrit; Hgb, hemoglobin; RBC, red blood cells; WBC, white blood cells.

**Table 2 vetsci-09-00549-t002:** Antibodies for flow cytometry (FC) analysis used in the case.

Antibody	Reactivity	Clone	Fluorochromes	Supplier
CD3	T cells	CA17.2A12	FITC	Bio-Rad
CD5	T cells	YKIX322.3	APC-eFluor780	Thermo Fisher Scientific
CD21	Mature B cells	CA2.1D6	PE	Bio-Rad
CD34	Hematopoietic stem cells	1H6	Alexa Fluor 405	R&D system

Bio-Rad (Hercules, CA, USA), Thermo Fisher Scientific (Waltham, MA, USA), R&D system (Minneapolis, MN, USA); FITC, fluorescein isothiocyanate; APC, allophycocyanin; PE, phycoerythrin.

## Data Availability

Not applicable.

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
