# Peer review of "Mott Cell Differentiation in Canine Multicentric B Cell Lymphoma with Cross-Lineage Rearrangement and Lineage Infidelity in a Dog"

_vetsci, 2022, doi:10.3390/vetsci9100549_

Round 1
Reviewer 1 Report
The manuscript describes an interesting case of lymphoma with Mott-cell differentiation in a mongrel dog. The diagnostic procedures and clinical decisions are well described. I have only few minor points:
- it looks like "lymphoblasts" and "lymphocytes" are used as synonims or occasionally confused in the manuscript. These two terms refer to 2 different maturation stages of lymphoid cells, with different morphological and phenotypic features. Please, carefully check the whole manuscript and correct. If necessary, you can use the term "lymphoid cells" when impossible to discriminate between the two entities.
- all cytological figures are poor quality and should be improved.
- figure4: please, add a morphological scattergram of both matrices and include details on gating strategy. Are the PB scattergams gated on lymphoid cells or on total nucleated cells?
- Table1: how was the antibody panel designed? Why were CD45 and CD8 not tested?
- Lines 151-153. The sentence "However...FNAC" is unclear. Cannot understand what the authors want to say here. Was PARR performed on PB sample, also? This was not stated in the manuscript.
- Lines 158-166. CD34 expression cannot be used as a stand-alone discriminator between lymphoma and acute leukemia, since both CD34+ LSA and CD34- acute leukemias have been reported. In the case presented, clinical finidngs (with localised solid lesions) support the diagnosis of LSA. This should be stated in the text.
- Lines 214-217. BM was not evaluated in the case presented. Presence of circulating neoplastic cells is not definitively diagnostic for bone marrow infiltration and authors should not evoke such a situation.
Author Response
Dear editor
We thank you for your time and consideration on our submission. Thus, it is with great pleasure that we resubmit our article for further consideration. We have incorporated changes that reflect the detailed suggestions you have graciously provided. Revisions made after carefully considering the comments of the reviewers and editor are as follows. The appropriate changes made in the revised manuscript are highlighted. We believe that these modifications have strengthened the manuscript and hope that the revised manuscript is suitable for publication in the Veterinary Sciences.
Comments and Suggestions for Authors
Reviewer 1: The manuscript describes an interesting case of lymphoma with Mott-cell differentiation in a mongrel dog. The diagnostic procedures and clinical decisions are well described. I have only few minor points:
Minor points
1-1 it looks like "lymphoblasts" and "lymphocytes" are used as synonims or occasionally confused in the manuscript. These two terms refer to 2 different maturation stages of lymphoid cells, with different morphological and phenotypic features. Please, carefully check the whole manuscript and correct. If necessary, you can use the term "lymphoid cells" when impossible to discriminate between the two entities.
Response: Thank you for your pointing this out. We have accordingly corrected the manuscript (line 80, 98, 110-111, 129, 154, 155, 189).
1-2 all cytological figures are poor quality and should be improved.
Response: Thank you for your pointing this out. We have accordingly corrected the figure (Case presentation section, line 84 and 127, page 3-4, Figure 1 and Figure 2).
1-3 figure4: please, add a morphological scattergram of both matrices and include details on gating strategy. Are the PB scattergams gated on lymphoid cells or on total nucleated cells?
Response: Thank you for your pointing this out. We have accordingly corrected the figure (Case presentation section, line 148, page 5, Figure 4). Please also refer to the Case presentation section: line 103-114, page 3. We added a description for the gating strategy in line with your comment.
1-4 Table1: how was the antibody panel designed? Why were CD45 and CD8 not tested?
Response: Thank you for your question. We have accordingly corrected the table (Case Presentation section, line 158, page 6, Table 2). We totally agreed with your question, however, unfortunately, because imports of those antibodies (CD45 and CD8) were impossible at that time because of covid-19. We thus could not test for CD8 and CD45 antibodies. Although the results would have been better if those antibodies had been tested, the diagnosis of B-cell LSA was made based on the other test results. Please also refer to the Discussion section: line 167-174, page 6.
1-5 Lines 151-153. The sentence "However...FNAC" is unclear. Cannot understand what the authors want to say here. Was PARR performed on PB sample, also? This was not stated in the manuscript.
Response: Thank you for your pointing this out. We have deleted the manuscript to reduce the confusion. PARR assay on peripheral blood was not performed.
1-6 Lines 158-166. CD34 expression cannot be used as a stand-alone discriminator between lymphoma and acute leukemia, since both CD34+ LSA and CD34- acute leukemias have been reported. In the case presented, clinical finidngs (with localised solid lesions) support the diagnosis of LSA. This should be stated in the text.
Response: Thank you for your pointing this out. We have revised the manuscript in line with your comment (Discussion section, line 179-190, page 6-7). Thanks a lot.
1-7 Lines 214-217. BM was not evaluated in the case presented. Presence of circulating neoplastic cells is not definitively diagnostic for bone marrow infiltration and authors should not evoke such a situation.
Response: Thank you for your pointing this out. We have revised the manuscript to reduce the confusion (Discussion section, line 243-247, page 8).

Reviewer 2 Report
The manuscript represents a case report on a dog with multicentric b-lymphoma associated with Mott cell differentiation, Cross-Lineage rearrangement, and Lineage Infidelity. Clinical data and response to CHOP-based chemotherapy were reported with a reduced survival time (81 days). The introduction and methodology of this paper are clearly described and the data are well presented. Although most of the references are up-to-date, the authors should add more references regarding canine Mott cell Lymphoma reports. Moreover, the Simple Summary is lacking and the authors should add it. The absence of histologic features is a limitation of the study, however, FC and PARR findings confirm the diagnosis. Since lymphoma is commonly diagnosed and the variable response to chemotherapy is a pivotal research topic in canine oncology, further studies in this field are needed. The English language and style are fine. In conclusion, I believe that this manuscript could be accepted for publication after some minor revisions.
Minor revisions:
Page 1 line 2. I suggest changing the title in order to emphasize that this is a case report.
Page 1 line 9. As stated in the new vetsci word layout, the authors should provide a concise “Simple Summary” during the submission. Please add it following the “instruction for authors” guidelines.
Page 1 line 23. To avoid redundancy, the authors should delete “…and this is the first report of MCL with CLRA and LI.” or rephrase and shorten the sentences in lines 22-24.
Page 1 line 32. The authors should add a reference at the end of the period.
Page 2 line 47. “is considered […] protocol for canine LSA”
Page 2 line 49. As the authors did in the discussions, clinical remission in response to chemotherapy should be added among the main prognostic factors. The authors should also add a reference at the end of the period.
Page 2 lines 55-57. The authors should change the order of the two sentences: “Since Mott cells are differentiated B cells, they can appear in B-cell LSA [9]. To the best of our knowledge, 10 reports of Mott cell differentiation in canine B-cell lymphoma (MCL) have been written to date”. Moreover should add the references regarding the specified works.
Page 2 line 58. Please rephrase “humanely destroyed” to make it more clear
Page 2 line 70. Was the lymph node measured? If yes, please add the dimensions.
Page 3 line 95 “…was found in…”
Page 5 lines 134-138. Please check the footer formatting
Page 6 line 159. Please, move the [11] reference to the end of the sentence.
Page 6 line 167. Please, rephrase the sentence avoiding “both”
Page 6 line 194. Is there any report of Mott cell differentiation associated with LI in humans?
Page 6 line 208. The authors should also add more recent papers as
Tamizharasan, S., et al. "Clinicopathological study of diffuse large B cell lymphoma in a Labrador dog: A case report." (2022);
Rimpo, Kenji, Miyuki Hirabayashi, and Aki Tanaka. "Lymphoma in Miniature Dachshunds: A retrospective multicenter study of 108 cases (2006‐2018) in Japan." Journal of Veterinary Internal Medicine (2022).
Page 6 lines 210-211. Please rephrase the sentence and spell out the abbreviation
Page 7 lines 228-229. Please, delete “in canines, MCL is extremely rare and exhibits an unusual clinical course” to avoid repetition.
Author Response
Minor revisions:
Page 1 line 2. I suggest changing the title in order to emphasize that this is a case report.
Response: Thank you for your pointing this out. We have accordingly corrected the title (line 4, page 1).
Page 1 line 9. As stated in the new vetsci word layout, the authors should provide a concise “Simple Summary” during the submission. Please add it following the “instruction for authors” guidelines.
Response: Thank you for your pointing this out. We have accordingly added the “Simple Summary” (line 10-22, page 1).
Page 1 line 23. To avoid redundancy, the authors should delete “…and this is the first report of MCL with CLRA and LI.” or rephrase and shorten the sentences in lines 22-24.
Response: Thank you for your pointing this out. We have accordingly deleted the manuscript (line 37, page 1).
Page 1 line 32. The authors should add a reference at the end of the period.
Response: Thank you for your pointing this out. We have accordingly corrected the manuscript (line 47, page 2; line 297, page 9).
Page 2 line 47. “is considered […] protocol for canine LSA”
Response: Thank you for your pointing this out. We have accordingly corrected the manuscript (line 64, page 2).
Page 2 line 49. As the authors did in the discussions, clinical remission in response to chemotherapy should be added among the main prognostic factors. The authors should also add a reference at the end of the period.
Response: Thank you for your pointing this out. We have accordingly corrected the manuscript (line 66-68 page 2; line 310-314, page 9).
Page 2 lines 55-57. The authors should change the order of the two sentences: “Since Mott cells are differentiated B cells, they can appear in B-cell LSA [9]. To the best of our knowledge, 10 reports of Mott cell differentiation in canine B-cell lymphoma (MCL) have been written to date”. Moreover should add the references regarding the specified works.
Response: Thank you for your pointing this out. We have accordingly corrected the manuscript (line 74-76, page 2; line 324-348, page 9-10).
Page 2 line 58. Please rephrase “humanely destroyed” to make it more clear
Response: Thank you for your pointing this out. We have accordingly corrected the manuscript to reduce the confusion (line 78, page 2).
Page 2 line 70. Was the lymph node measured? If yes, please add the dimensions.
Response: Thank you for your pointing this out. We have accordingly corrected the manuscript (line 90, page 2).
Page 3 line 95 “…was found in…”
Response: Thank you for your pointing this out. We have accordingly corrected the manuscript (line 143, page 5).
Page 5 lines 134-138. Please check the footer formatting
Response: Thank you for your pointing this out. We have accordingly corrected the footer (line 163-165, page 5).
Page 6 line 159. Please, move the [11] reference to the end of the sentence.
Response: Thank you for your pointing this out. We have accordingly corrected the manuscript (line 200, page 7).
Page 6 line 167. Please, rephrase the sentence avoiding “both”
Response: Thank you for your pointing this out. We have deleted “both Mott cell differentiation and” (line 209, page 7).
Page 6 line 194. Is there any report of Mott cell differentiation associated with LI in humans?
Response: Thank you for your question. To the best of our knowledge, there are no reports of Mott cell differentiation associated with LI in humans. We regret that we could not include a comparison and description of the relevant human literature with this report.
Page 6 line 208. The authors should also add more recent papers as
Tamizharasan, S., et al. "Clinicopathological study of diffuse large B cell lymphoma in a Labrador dog: A case report." (2022);
Rimpo, Kenji, Miyuki Hirabayashi, and Aki Tanaka. "Lymphoma in Miniature Dachshunds: A retrospective multicenter study of 108 cases (2006‐2018) in Japan." Journal of Veterinary Internal Medicine (2022).
Response: Thank you for your pointing this out. We have accordingly corrected the manuscript (line 244, page 8; line 343-348, page 10).
Page 6 lines 210-211. Please rephrase the sentence and spell out the abbreviation
Response: Thank you for your pointing this out. We have accordingly corrected the manuscript (line 255-256, page 8).
Page 7 lines 228-229. Please, delete “in canines, MCL is extremely rare and exhibits an unusual clinical course” to avoid repetition.
Response: Thank you for your pointing this out. We have accordingly deleted the manuscript (line 275, page 8).

Reviewer 3 Report
Dear Authors,
In their work Kim and co-authors described the case study of canine lymphoma entitled: Mott Cell Differentiation in Canine Multicentric B Cell Lymphoma with Cross-Lineage Rearrangement and Lineage Infidelity.
In Veterinary Sciences authors guide: Case reports present detailed information on the symptoms, signs, diagnosis, treatment (including all types of interventions), and outcomes of an individual patient. Case reports usually describe new or uncommon conditions that serve to enhance medical care or highlight diagnostic approaches. And …..“Figures, Schemes and Tables should be inserted into the main text close to their first citation and must be numbered”
And thus, in my opinion paragraph 2 - Case presentation needs major changes.
Major Comments:
My main comment would be regarding the phenotyping strategy, which is crucial diagnostic element of this work. It is necessary to show the gating strategy to explain the phenotype of cells from LN CD3+/CD4-/CD5-/CD21+/CD34- and lymphocytes from blood CD3-/CD4-/CD5-/CD21+/CD34- - line 103
Line 127 Figure 4 is not readable enough as gating strategy.
Discussion paragraph
In that paragraph authors should also discuss their results with the results of other case studies:
-De Zan G, Zappulli V, Cavicchioli L, Di Martino L, Ros E, Conforto G, Castagnaro M. Gastric B-cell lymphoma with Mott cell differentiation in a dog. J Vet Diagn Invest. 2009 Sep;21(5):715-9. doi: 10.1177/104063870902100521. PMID: 19737772.
-Kol A, Christopher MM, Skorupski KA, Tokarz D, Vernau W. B-cell lymphoma with plasmacytoid differentiation, atypical cytoplasmic inclusions, and secondary leukemia in a dog. Vet Clin Pathol. 2013 Mar;42(1):40-6. doi: 10.1111/vcp.12003. Epub 2012 Dec 3. PMID: 23205858.
-Stacy NI, Nabity MB, Hackendahl N, Buote M, Ward J, Ginn PE, Vernau W, Clapp WL, Harvey JW. B-cell lymphoma with Mott cell differentiation in two young adult dogs. Vet Clin Pathol. 2009 Mar;38(1):113-20. doi: 10.1111/j.1939-165X.2008.00101.x. Epub 2008 Dec 18. PMID: 19171017.
Line 128. Information about Figure 4-is missing in main text;
The paragraph “introduction” is well written, but in my opinion epidemiological aspect of LSA should be discussed there. Moreover, authors noticed that only 10 articles describe the Mot Cell differentiation in veterinary medicine – line 55 but there is no references in literature section.
Minor comments
Line 72- all results of CBC should be presented;
Line 76- x400 should be changed into x 400 objective;
Line 76 in figure 1A Mott cells are arrowheads but in figure 1B are arrows- it should be unified in both figures; Mott cells in blood smear should be presented in standard field of view;
Line 86. Figure 2B- what the arrow shows? Fig2C what stars show?
Line 91- change reflectometer into refractometer
Line 133- Table1 conjugates should be changed into fluorochromes
The paper represents clinically valuable scientific work, and it can be accepted for publication in Veterinary Sciences after major revision.

Author Response
Dear editor
We thank you for your time and consideration on our submission. Thus, it is with great pleasure that we resubmit our article for further consideration. We have incorporated changes that reflect the detailed suggestions you have graciously provided. Revisions made after carefully considering the comments of the reviewers and editor are as follows. The appropriate changes made in the revised manuscript are highlighted. We believe that these modifications have strengthened the manuscript and hope that the revised manuscript is suitable for publication in the Veterinary Sciences.
Reviewer 2: In Veterinary Sciences authors guide: Case reports present detailed information on the symptoms, signs, diagnosis, treatment (including all types of interventions), and outcomes of an individual patient. Case reports usually describe new or uncommon conditions that serve to enhance medical care or highlight diagnostic approaches. And …..“Figures, Schemes and Tables should be inserted into the main text close to their first citation and must be numbered”
And thus, in my opinion paragraph 2 - Case presentation needs major changes.
Major Comments:
2-1 My main comment would be regarding the phenotyping strategy, which is crucial diagnostic element of this work. It is necessary to show the gating strategy to explain the phenotype of cells from LN CD3+/CD4-/CD5-/CD21+/CD34- and lymphocytes from blood CD3-/CD4-/CD5-/CD21+/CD34- - line 103
Line 127 Figure 4 is not readable enough as gating strategy.
Response: Thank you for your pointing this out. We added a description for the gating strategy on our revised manuscript (Case presentation section: line 103-114, page 3). Please also refer to answer of 1-3 (rebuttal letter).
2-2 Discussion paragraph
In that paragraph authors should also discuss their results with the results of other case studies:
-De Zan G, Zappulli V, Cavicchioli L, Di Martino L, Ros E, Conforto G, Castagnaro M. Gastric B-cell lymphoma with Mott cell differentiation in a dog. J Vet Diagn Invest. 2009 Sep;21(5):715-9. doi: 10.1177/104063870902100521. PMID: 19737772.
-Kol A, Christopher MM, Skorupski KA, Tokarz D, Vernau W. B-cell lymphoma with plasmacytoid differentiation, atypical cytoplasmic inclusions, and secondary leukemia in a dog. Vet Clin Pathol. 2013 Mar;42(1):40-6. doi: 10.1111/vcp.12003. Epub 2012 Dec 3. PMID: 23205858.
-Stacy NI, Nabity MB, Hackendahl N, Buote M, Ward J, Ginn PE, Vernau W, Clapp WL, Harvey JW. B-cell lymphoma with Mott cell differentiation in two young adult dogs. Vet Clin Pathol. 2009 Mar;38(1):113-20. doi: 10.1111/j.1939-165X.2008.00101.x. Epub 2008 Dec 18. PMID: 19171017.
Response: Thank you for your pointing this out. We have accordingly corrected the manuscript (Discussion section, line 227-235, page 7).
2-3 Line 128. Information about Figure 4-is missing in main text;
Response: Thank you for your pointing this out. We have accordingly corrected the manuscript (Case presentation section, line 109 and 113, page 3).
2-4 The paragraph “introduction” is well written, but in my opinion epidemiological aspect of LSA should be discussed there. Moreover, authors noticed that only 10 articles describe the Mot Cell differentiation in veterinary medicine – line 55 but there is no references in literature section.
Response: Thank you for your pointing this out. The manuscript was revised accordingly (line 32-37, page 1; line 60-61, page 2; line 297-315, page 9).
Minor Comments:
Line 72- all results of CBC should be presented;
Response: Thank you for your pointing this out. We have accordingly corrected the manuscript. We added a table to the revised manuscript(Case Presentation section, line 77-80, page 2, Table 1).
Line 76- x400 should be changed into x 400 objective;
Response: Thank you for your pointing this out. We have accordingly corrected the manuscript (Case Presentation section, line 85, page 3, Figure 1).
Line 76 in figure 1A Mott cells are arrowheads but in figure 1B are arrows- it should be unified in both figures; Mott cells in blood smear should be presented in standard field of view;
Response: Thank you for your pointing this out. We have accordingly corrected the figure to reduce the confusion (Case Report section, line 86 and 88, page 3, Figure 1).
Line 86. Figure 2B- what the arrow shows? Fig2C what stars show?
Response: Thank you for your question. We have corrected the manuscript to reduce the confusion (Case Presentation section, line 124-126, page 4, Figure 2).
Line 91- change reflectometer into refractometer
Response: Thank you for your pointing this out. We have accordingly corrected the manuscript (Case Presentation section, line 97, page 3).
Line 133- Table1 conjugates should be changed into fluorochromes
Response: Thank you for your pointing this out. We have accordingly corrected the manuscript (Case Presentation section, line 158, page 6, Table 2).

Round 2
Reviewer 3 Report
Dear Authors,
My main comment still would be regarding the phenotyping strategy, which is crucial diagnostic element of this work.
The gating strategy is not hierarchical. After choosing region of lymphocytes, doublets should be removed (FSC-A FSC-H). Next only singlets should be analyzed based on CDs expression;
The panel of CD molecules is chosen properly, however small diversity of used fluorochromes made it impossible to show in LN cells: CD3+/CD4-/CD5-/CD21+/CD34- as well as in blood cells: CD3-/CD4-/CD5-/CD21+/CD34- in hierarchical way;
To prove the lineage infidelity (IL) in LN sample, cells should be labelled in one tube with CD5APC, CD21RPE and CD34FITC
Uploaded gating strategy in figure 4 is still unclear and contains mistakes:
- There is no dot plot histogram showing CD4 expression for LN and whole blood sample, and thus CD4- should be removed from the phenotype
Figure 4.A- the phenotype of lymphocytes from LN is: CD3+/CD4-/CD5-/CD21+/CD34-, but in CD3/CD5 dot plot quadrant 2 (Q2) is a quadrant with double positive cells (CD3+CD5+) and therefore the chosen region should be only Q3; this changes the % values of gated cells.
Author Response
The gating strategy is not hierarchical. After choosing region of lymphocytes, doublets should be removed (FSC-A FSC-H). Next only singlets should be analyzed based on CDs expression;
Response: Thank you for your pointing this out. We have accordingly corrected the manuscript (line 121-123, page 4; line 168,171-172 page 6).
The panel of CD molecules is chosen properly, however small diversity of used fluorochromes made it impossible to show in LN cells: CD3+/CD4-/CD5-/CD21+/CD34- as well as in blood cells: CD3-/CD4-/CD5-/CD21+/CD34- in hierarchical way;
Response: Thank you for your pointing this out. We have accordingly corrected the table 2 (line 162, page 5).
To prove the lineage infidelity (IL) in LN sample, cells should be labelled in one tube with CD5APC, CD21RPE and CD34FITC
Response: Thank you for your pointing this out. Since our patient was positive for CD3 and CD21, we think you may be describing about CD3 rather than CD5. We totally agreed with your opinion. We think that it would have been clearer for representing LI if there were CD3 and CD21 dot plots as you mentioned, however, unfortunately, the FC test was firstly conducted for diagnostic purposes at that time and we could not first suspect the possibility of LI. We are very sorry again that we could not provide high-quality, all-inclusive data. However, all cells were labelled in one tube with CD5 APC, CD21 PE and CD34 Alexa Fluor 405. Since dot plots using two parameters were technically available, all dot plots could not be described even though CD3, CD21 and CD34 were all labeled in one LN sample. Please refer to line 118-129 and 162-175 of revised manuscript.
Uploaded gating strategy in figure 4 is still unclear and contains mistakes:
- There is no dot plot histogram showing CD4 expression for LN and whole blood sample, and thus CD4- should be removed from the phenotype
Response: Thank you for your pointing this out. We have accordingly revised the manuscript (line 124,126,162, page 4,5).
Figure 4.A- the phenotype of lymphocytes from LN is: CD3+/CD4-/CD5-/CD21+/CD34-, but in CD3/CD5 dot plot quadrant 2 (Q2) is a quadrant with double positive cells (CD3+CD5+) and therefore the chosen region should be only Q3; this changes the % values of gated cells.
Response: Thank you for your pointing this out. We have accordingly corrected the figure 4.A (line 168, page 6).

Round 3
Reviewer 3 Report
Dear authors
After major revisions, paper represents good scientific quality, and it can be accepted for the publication in its current version.